# Fault Estimation Method for Nonlinear Time-Delay System Based on Intermediate Observer-Application on Quadrotor Unmanned Aerial Vehicle

**DOI:** 10.3390/s23010034

**Published:** 2022-12-20

**Authors:** Qingnan Huang, Jingru Qi, Xisheng Dai, Qiqi Wu, Xianming Xie, Enze Zhang

**Affiliations:** School of Automation, Guangxi University of Science and Technology, Liuzhou 545006, China

**Keywords:** fault estimation, intermediate observer, quadrotor UAV, actuator fault, sensor fault

## Abstract

In this paper, the problem of actuator and sensor faults of a quadrotor unmanned aerial vehicle (QUAV) system is studied. In the system fault model, time delay, nonlinear term, and disturbances of QUAV during the flight are considered. A fault estimation algorithm based on an intermediate observer is proposed. To deal with a single actuator fault, an intermediate variable is introduced, and the intermediate observer is designed for the system to estimate fault. For simultaneous actuator and sensor faults, the system is first augmented, and then two intermediate variables are introduced, and an intermediate observer is designed for the augmented system to estimate the system state, faults, and disturbances. The Lyapunov–Krasovskii functional is used to prove that the estimation error system is uniformly eventually bounded. The simulation results verify the feasibility and effectiveness of the proposed fault estimation method.

## 1. Introduction

QUAV is a special type of UAV system equipped with four propellers. It has unique advantages such as flexible control, small size, vertical take-off and landing, and strong adaptability. QUAVs have played a very important role in personnel rescue, military surveillance, vegetation protection, pesticide spraying, etc. [1,2,3,4,5]. However, with the improvement of system performance, the control system will become more complex. In the process of frequently performing tasks, various faults sometimes occur. The occurrence of QUAV system faults not only causes serious damage to the UAV itself but also poses a serious threat to human and environmental safety [6,7,8,9]. Therefore, as far as the current application of QUAV is concerned, strengthening the fault diagnosis and fault-tolerant control (FTC) of the system can effectively broaden the application field of quadrotor UAV and ensure the stability and safety of the system [10,11,12,13].

In terms of the control systems, fault diagnosis is significant, many scholars have devoted considerable attention to the research of fault diagnosis methods, which have made rapid development and achieved many theoretical results. These results have been successfully applied in various fields of industry, such as UAVs, high-speed trains, rolling bearings chemical plants, etc. At the same time, new theories and technologies are still emerging. According to traditional classification methods, model-based, data-based, and knowledge-based methods are three types of fault diagnosis methods [14,15,16]. In the field of QUAVs, with the increasing demand for system reliability, safety, and stability, the research on UAV fault diagnosis has progressed rapidly, among which actuator fault diagnosis has achieved the most results.

There are many methods for the actuator fault of QUAV. For example, Freddi et al. proposed a Thau observer-based fault diagnosis method for nonlinear systems, which can be used to detect an actuator fault or sensor fault, but their method is not suitable for fault isolation and estimation, and is only for UAV models [17]. For the QUAV attitude control system, the ref. [18] proposed a joint observer method, by decoupling the original system into two subsystems, where an adaptive observer and sliding mode observer were designed for the two subsystems, respectively, and achieved satisfactory results. For the QUAV system with external disturbance, Avram et al. introduced a nonlinear adaptive estimation technique to realize the detection, isolation, and adjustment of actuator fault [19]. Cen et al. proposed a new adaptive Thau observer that optimizes a robust fault diagnosis scheme and implements it on an actual QUAV [20]. In order to obtain the estimation of QUAV fault, an adaptive observer-based fault estimation algorithm was designed in ref. [21] and achieved good results.

Although extensive research efforts on the actuator fault of quadrotor UAVs, the existing research does not involve sensor fault diagnosis. Many approaches have been proposed in order to deal with the sensor fault of the QUAV. In view of the sensor fault in the measurement of the accelerometer and gyroscope of the QUAV, a sliding mode observer method was proposed in ref. [22] to estimate the roll and pitch angles of the UAV, and a nonlinear fault diagnosis algorithm to estimate sensor fault. Ref. [23] proposed a method for the detection and isolation of the sensor fault by designing an unknown input observer. However, this method is unable to estimate the sensor fault. For the nonlinear dynamic model of quadrotor UAV, a new scheme based on a neural network observer was designed in ref. [24] for UAV system sensor fault detection and isolation, this method can quickly detect sensor fault.

In addition to a single fault, the system may also have a simultaneous actuator fault and sensor fault. Therefore, there are many methods to cope with this condition. When actuator fault and sensor fault occur at the same time and the system contains unknown disturbance, the ref. [25] can achieve the function of estimating sensor fault by designing a robust sliding mode observer, but it can only be used to detect actuator fault and cannot estimate actuator fault. Ref. [26] proposed a new reduced-order sliding-mode observer, which can estimate the system state and faults, and achieve good estimation results. When the matching condition of the observer is not satisfied, ref. [27] first augmented the system and introduced a new state vector, then used an adaptive robust sliding mode observer to detect actuator and sensor faults. For nonlinear systems, an intermediate observer was proposed in ref. [28] for the first time, which broke through the constraints of the observer matching condition. Not only the single actuator fault can be estimated well, but also the actuator fault and sensor fault can be estimated effectively at the same time. However this paper ignores the influence of external disturbance, and in the design of the intermediate observer, the measurement output is not considered. In view of the problems in ref. [28], ref. [29] improved the results by considering the full measurement output in the design process of the state estimation observer. However, this design is not suitable for the case where the actuator and sensor fail at the same time, and it can only estimate the actuator fault. For a class of switched fuzzy systems, ref. [30] proposed a new switched fuzzy observer method by augmenting the system, which can simultaneously estimate actuator fault and sensor fault. It is shown that this method has a good estimation effect. In ref. [31], a special interval observer based on the zonotope method is proposed for the T-S fuzzy system, however, this method is only suitable for discrete-time systems, not for continuous-time systems.

In the actual quadrotor UAV system, the actuator and sensor may fail at the same time. In addition, the actual system will also have a time delay, which will sometimes affect the system’s stability. Based on the above reasons, a fault estimation strategy based on the intermediate observer is proposed in this paper to solve the fault estimation problem of a QUAV system with time delay and disturbances. The main contributions of this paper are as follows:Different from the system model used in references [18,28], this paper considers the problem of a single actuator fault and simultaneous actuator and sensor faults with time delay.For a single actuator fault, the effect of time delay is considered, an intermediate variable is introduced, and the intermediate observer is designed to estimate the state of the system and the actuator fault.When actuator fault and sensor fault occur at the same time, we consider the influence of time delay, unknown input, and measurement noise disturbances in order to facilitate the handling of a sensor fault. The original system is augmented first, two intermediate variables are introduced, and the intermediate observer is designed for the augmented system, which is used to estimate the system state, actuator fault, sensor fault, and disturbances.

**Notations.** 
*Some notations will be used in this article. Define $R^{n}$ as the n-dimensional Euclidean space. For a matrix A, $A^{T}$ and $A^{-1}$ represent its transpose and inverse. Matrix I is an identity matrix of appropriate dimensions. $∥\cdot ∥$ denotes the Euclidean norm of vectors or matrices. For a symmetric matrix A, A>0 means that the matrix is positive definite. The symbol * in the matrix represents the symmetric term. $\lambda_{\min}\left(A\right)$ represents the smallest eigenvalue of the matrix A.*


## 2. Design of the Intermediate Observer

### 2.1. Observer Design for Actuator Fault Diagnosis

The quadrotor UAV is an under-actuated system, which has six degrees of freedom but only four actual inputs. According to a large number of studies by scholars on the modeling of quadrotor UAVs, for the convenience of modeling, the following assumptions need to be made without loss of generality. First, the aircraft is a rigid body and the quality of the aircraft has a uniform distribution. Second, the aircraft’s lift surface and centre of gravity are in the same plane. Based on the above assumptions, ref. [18] established the following dynamics model of the quadrotor UAV attitude system:(1)$$\left\{\begin{array}{c} J_{x}\ddot{\phi }=\left(J_{y}-J_{z}\right)\dot{\theta }\dot{\psi }+lK_{lc}\left(V_{r}-V_{l}\right)-K_{afcx}{\dot{\phi }}^{2}\\ J_{y}\ddot{\theta }=\left(J_{z}-J_{x}\right)\dot{\psi }\dot{\phi }+lK_{lc}\left(V_{f}-V_{b}\right)-K_{afcy}{\dot{\theta }}^{2}\\ J_{z}\ddot{\psi }=\left(J_{x}-J_{y}\right)\dot{\theta }\dot{\phi }+K_{vc}\left(V_{f}+V_{b}\right)+K_{vn}\left(V_{r}+V_{l}\right)-K_{afcz}{\dot{\psi }}^{2} \end{array}\right.$$
where $J_{x}$, $J_{y}$, $J_{z}$ are the rotational inertia of the roll axis, pitch axis, and yaw axis, and the Euler angles of the body axes are $\Pi ={\left[\phi ,\theta ,\psi \right]}^{T}$ that refer to roll, pitch, and yaw angles, respectively. $V_{f}$, $V_{b}$, $V_{r}$, $V_{l}$ are the voltage of the front, rear, right, and left motors. *l* is the distance between the gravity centre of the quadrotor and the motor rotation axis, $K_{lc}$ represents the propeller force-thrust constant, $K_{vc}$ and $K_{vn}$ are the counter and normal rotation propeller torque–thrust constant. $K_{afcx}$, $K_{afcy}$, and $K_{afcz}$ are the drag coefficients of the corresponding axis.

By defining the state vector $x\left(t\right)={\left[\phi \;\dot{\phi }\;\theta \;\dot{\theta }\;\psi \;\dot{\psi }\right]}^{T}$, output vector $y\left(t\right)={\left[\phi \;\theta \;\psi \;\right]}^{T}$, and control input vector $u\left(t\right)={\left[V_{f}\;V_{b}\;V_{r}\;V_{l}\right]}^{T}$, the quadrotor system is written in matrix form and can be expressed as
(2)$$\left\{\begin{array}{c} \dot{x}\left(t\right)=Ax\left(t\right)+Bu\left(t\right)+g\left(t,x\left(t\right)\right)\\ y\left(t\right)=Cx\left(t\right) \end{array}\right.$$

Considering the case of a single actuator fault in a quadrotor UAV, in practical applications, the system will also have a time delay. Therefore, the following quadrotor UAV model with actuator fault and time delay is established:(3)$$\left\{\begin{array}{c} \;\dot{x}\left(t\right)=Ax\left(t\right)+A_{z}x\left(t-z\right)+g\left(t,x\left(t\right)\right)+g_{z}\left(t,x\left(t-z\right)\right)+Bu\left(t\right)+Ef_{a}\left(t\right)\\ y\left(t\right)=Cx\left(t\right) \end{array}\right.$$
where $x\left(t\right)\in R^{n}$ is the system state vector, $u\left(t\right)\in R^{m}$ donates the system input, and $y\left(t\right)\in R^{l}$ is the system output vector. $A,A_{z},B,C$, and *E* are real constant matrices of appropriate dimensions. *E* is full column rank. (A,B) and (A,C) are assumed to be controllable and observable, respectively. $f_{a}\left(t\right)\in R^{r}$ represents the actuator fault, z∈R is the time delay.

Some assumptions are as follows:

**Assumption** **1**.
*The actuator faults $f_{a}\left(t\right)$ is unknown time-varying and satisfies $∥{\dot{f}}_{a}\left(t\right)∥\leq \alpha$ with α≥0, where α is a positive number.*


**Assumption** **2.**
*The nonlinear vectors $g\left(t,x\left(t\right)\right)$ and $g\left(t,x\left(t-z\right)\right)$ are assumed to be known and meet the Lipschitz condition about $x\left(t\right)\in R^{n}$ and $x\left(t-z\right)\in R^{n}$, $∥g\left(t,x\left(t\right)\right)-g\left(t,\hat{x}\left(t\right)\right)∥\leq l_{1}∥x\left(t\right)-\hat{x}\left(t\right)∥$, $∥g_{z}\left(t,x\left(t-z\right)\right)-g_{z}\left(t,\hat{x}\left(t-z\right)\right)∥\leq l_{2}∥x\left(t-z\right)-\hat{x}\left(t-z\right)∥$.*


**Assumption** **3.**
*For every complex number λ with a non-negative real part, the following equalities hold:*

(4)
$$rank\left[\begin{array}{cc} A-\lambda I & E\\ C & 0 \end{array}\right]=n+rank\left(E\right)$$



**Remark** **1.**
*According to ref. [28], assumption 1 means that α could be unknown, and this assumption is more general. Assumption 2 is common because many actual nonlinear systems are Lipschitz.*


**Remark** **2.**
*The intermediate observer used in this paper only needs to satisfy the condition that E is full column rank, which overcomes the restriction of the observer matching condition rank(CE)=rank(E). Assumption 3 guarantees that the invariant zero points of the system (A,C,E) are in the left half of the complex plane, and is common in the literature on fault estimation.*


**Lemma** **1.**
*For any vector $X,Y\in R^{n}$, scalar ε>0, and positive definite matrix P, it holds that [32]:*

(5)
$$2X^{T}PY\leq \varepsilon X^{T}PX+\frac{1}{\varepsilon }Y^{T}PY$$



To design an intermediate observer for the nonlinear system (3), an intermediate variable is first introduced.
(6)$$\tau \left(t\right)=f_{a}\left(t\right)-\sigma E^{T}x\left(t\right)$$

From Equations (3) and (6), it can be obtained that
(7)$$\begin{array}{cc} \dot{\tau }\left(t\right)= & {\dot{f}}_{a}\left(t\right)-\sigma E^{T}\left[Ax\left(t\right)+A_{z}x\left(t-z\right)+Bu\left(t\right)\right.\\ \; & +g\left(t,x\left(t\right)\right)+g_{z}\left(t,x\left(t-z\right)\right)\left.+Ef_{a}\left(t\right)\right] \end{array}$$

Based on Equations (3) and (7), a fault estimation observer is proposed as follows:(8)$$\dot{\hat{x}}\left(t\right)=A\hat{x}\left(t\right)+A_{z}\hat{x}\left(t-z\right)+Bu\left(t\right)+\hat{g}\left(t,x\left(t\right)\right)+{\hat{g}}_{z}\left(t,x\left(t-z\right)\right)+E{\hat{f}}_{a}\left(t\right)+F\left(y-\hat{y}\right)$$
(9)$$\begin{array}{cc} \dot{\hat{\tau }}\left(t\right)= & -\sigma E^{T}E\hat{\tau }\left(t\right)-\sigma E^{T}\left[A\hat{x}\left(t\right)+A_{z}\hat{x}\left(t-z\right)\right.\\ \; & +Bu\left(t\right)+\hat{g}\left(t,x\left(t\right)\right)+{\hat{g}}_{z}\left(t,x\left(t-z\right)\right)+\left.\sigma EE^{T}\hat{x}\left(t\right)\right.] \end{array}$$
(10)$$\hat{y}\left(t\right)=C\hat{x}\left(t\right)$$
(11)$${\hat{f}}_{a}\left(t\right)=\hat{\tau }\left(t\right)+\sigma E^{T}\hat{x}\left(t\right)$$
where $\hat{x}\left(t\right)$, $\hat{y}\left(t\right)$, ${\hat{f}}_{a}\left(t\right)$, and $\hat{\tau }\left(t\right)$ represent the estimation of $x\left(t\right)$, $y\left(t\right)$, $f_{a}\left(t\right)$, and $\tau \left(t\right)$, respectively. Define the estimation error $e_{x}\left(t\right)=x\left(t\right)-\hat{x}\left(t\right)$, $e_{x}\left(t-z\right)=x\left(t-z\right)-\hat{x}\left(t-z\right)$, $e_{\tau }\left(t\right)=\tau \left(t\right)-\hat{\tau }\left(t\right)$, and $e_{f_{a}}\left(t\right)=f_{a}\left(t\right)-{\hat{f}}_{a}\left(t\right)$, then
(12)$${\dot{e}}_{x}\left(t\right)=\left(A-FC+\sigma EE^{T}\right)e_{x}\left(t\right)+A_{z}e_{x}\left(t-z\right)+G+G_{z}+Ee_{\tau }\left(t\right)$$
(13)$${\dot{e}}_{\tau }\left(t\right)={\dot{f}}_{a}\left(t\right)-\sigma E^{T}Ee_{\tau }\left(t\right)-\sigma E^{T}\left[\left(A+\sigma EE^{T}\right)e_{x}\left(t\right)+A_{z}e_{x}\left(t-z\right)+G+G_{z}\right]$$
where $G=g\left(t,x\left(t\right)\right)-\hat{g}\left(t,x\left(t\right)\right)$, $G_{z}=g_{z}\left(t,x\left(t-z\right)\right)-{\hat{g}}_{z}\left(t,x\left(t-z\right)\right)$.

**Theorem** **1.**
*Under the above assumptions, for the given positive constants σ, ε, if there exist scalars δ>0 and matrices P>0, H such that*

(14)
$$\Omega =\left[\begin{array}{ccccccc} \Omega_{11} & \Omega_{12} & \Omega_{13} & l_{1}P & \sigma \delta l_{1}I & 0 & 0\\ {}* & \Omega_{22} & \Omega_{23} & 0 & 0 & l_{2}P & \sigma \delta l_{2}I\\ {}* & * & \Omega_{33} & 0 & 0 & 0 & 0\\ {}* & * & * & -\varepsilon I & 0 & 0 & 0\\ {}* & * & * & * & -\varepsilon I & 0 & 0\\ {}* & * & * & * & * & -\varepsilon I & 0\\ {}* & * & * & * & * & * & -\varepsilon I \end{array}\right]<0$$

*the estimation error system (12) and (13) is guaranteed to be uniformly ultimately bounded, and the estimator gain is given by $F=P^{-1}H$, where $\Omega_{11}=PA-HC+A^{T}P-C^{T}H^{T}+\mu PEE^{T}+\mu EE^{T}P+2\varepsilon I+Q$, $\Omega_{12}=PA_{z}$, $\Omega_{13}=PE-\sigma \delta A^{T}E-\sigma^{2}\delta EE^{T}E$, $\Omega_{22}=-Q$, $\Omega_{23}=-\sigma \delta A_{z}^{T}E$, $\Omega_{33}=-2\sigma \delta E^{T}E+\frac{1}{\varepsilon }I+2\varepsilon E^{T}E$.*


**Proof.** Consider the Lyapunov–Krasovskii functional as
(15)$$V=V_{1}+V_{2}+V_{3}$$
where
$$\begin{array}{cc} \; & V_{1}=e_{x}^{T}\left(t\right)Pe_{x}\left(t\right),V_{2}=e_{\tau }^{T}\left(t\right)\Gamma e_{\tau }\left(t\right),\\ \; & V_{3}=\int_{t-z}^{t}e_{x}^{T}\left(s\right)Qe_{x}\left(s\right)ds,\Gamma =\delta I. \end{array}$$Thus,
(16)$$\begin{array}{cc} {\dot{V}}_{1}= & e_{x}^{T}\left(t\right)\left[P\left(A-FC\right)+{\left(A-FC\right)}^{T}P\right]e_{x}\left(t\right)+2\sigma e_{x}^{T}\left(t\right)PEE^{T}e_{x}\left(t\right)+2e_{x}^{T}\left(t\right)PEe_{\tau }\left(t\right)\\ \; & +2e_{x}^{T}\left(t\right)PA_{z}e_{x}\left(t-z\right)+2e_{x}^{T}\left(t\right)PG+2e_{x}^{T}\left(t\right)PG_{z} \end{array}$$
(17)$$\begin{array}{cc} {\dot{V}}_{2}= & 2\delta e_{\tau }^{T}\left(t\right){\dot{f}}_{a}\left(t\right)-2\sigma \delta e_{\tau }^{T}\left(t\right)E^{T}Ee_{\tau }\left(t\right)-2\sigma \delta e_{\tau }^{T}\left(t\right)E^{T}\left(A+\sigma EE^{T}\right)e_{x}\left(t\right)\\ \; & -2\sigma \delta e_{\tau }^{T}\left(t\right)E^{T}A_{z}e_{x}\left(t-z\right)-2\sigma \delta e_{\tau }^{T}\left(t\right)E^{T}G-2\sigma \delta e_{\tau }^{T}\left(t\right)E^{T}G_{z} \end{array}$$
(18)$${\dot{V}}_{3}=e_{x}^{T}\left(t\right)Qe_{x}\left(t\right)-e_{x}^{T}\left(t-z\right)Qe_{x}\left(t-z\right)$$According to Lemma 1, the following inequalities are always true:
(19)$$2e_{x}^{T}\left(t\right)PG\leq \varepsilon e_{x}^{T}\left(t\right)e_{x}\left(t\right)+\frac{1}{\varepsilon }l_{1}^{2}e_{x}^{T}\left(t\right)P^{2}e_{x}\left(t\right)$$
(20)$$2e_{x}^{T}\left(t\right)PG_{z}\leq \varepsilon e_{x}^{T}\left(t\right)e_{x}\left(t\right)+\frac{1}{\varepsilon }l_{2}^{2}e_{x}^{T}\left(t-z\right)P^{2}e_{x}\left(t-z\right)$$
(21)$$2\delta e_{\tau }^{T}\left(t\right){\dot{f}}_{a}\left(t\right)\leq \frac{1}{\varepsilon }e_{\tau }^{T}\left(t\right)e_{\tau }\left(t\right)+\varepsilon \delta^{2}\alpha^{2}$$
(22)$$-2\sigma \delta e_{\tau }^{T}\left(t\right)E^{T}G\leq \varepsilon e_{\tau }^{T}\left(t\right)E^{T}Ee_{\tau }\left(t\right)+\frac{1}{\varepsilon }l_{1}^{2}\delta^{2}\sigma^{2}e_{x}^{T}\left(t\right)e_{x}\left(t\right)$$
(23)$$-2\sigma \delta e_{\tau }^{T}\left(t\right)E^{T}G_{z}\leq \frac{1}{\varepsilon }l_{2}^{2}\delta^{2}\sigma^{2}e_{x}^{T}\left(t-z\right)e_{x}\left(t-z\right)+\varepsilon e_{\tau }^{T}\left(t\right)E^{T}Ee_{\tau }\left(t\right)$$Define $e\left(t\right)={\left[e_{x}^{T}\left(t\right)\;e_{x}^{T}\left(t-z\right)\;e_{\tau }^{T}\left(t\right)\right]}^{T}$, then, it can be obtained that
(24)$$\dot{V}=e^{T}\Sigma e+\kappa$$
where
$$\begin{array}{c} \Sigma =\left[\begin{array}{ccc} \Sigma_{11} & \Sigma_{12} & \Sigma_{13}\\ {}* & \Sigma_{22} & \Sigma_{23}\\ {}* & * & \Sigma_{33} \end{array}\right] \end{array}$$
and
$$\Sigma_{11}=P\left(A-FC+\sigma EE^{T}\right)+{\left(A-FC+\sigma EE^{T}\right)}^{T}P+2\varepsilon I+Q+\frac{1}{\varepsilon }l_{1}^{2}P^{2}+\frac{1}{\varepsilon }\sigma^{2}\delta^{2}l_{1}^{2}I$$
$$\begin{array}{cc} \; & \Sigma_{12}=PA_{z}\\ \; & \Sigma_{13}=PE-\sigma \delta A^{T}E-\sigma^{2}\delta EE^{T}E\\ \; & \Sigma_{22}=-Q+\frac{1}{\varepsilon }l_{2}^{2}P^{2}+\frac{1}{\varepsilon }\sigma^{2}\delta^{2}l_{2}^{2}I\\ \; & \Sigma_{23}=-\sigma \delta A_{z}^{T}E\\ \; & \Sigma_{33}=-2\sigma \delta E^{T}E+\frac{1}{\varepsilon }I+2\varepsilon E^{T}E\\ \; & \kappa =\varepsilon \delta^{2}\alpha^{2} \end{array}$$According to Schur’s complement theorem, Σ<0 is equivalent to Equation (Equation 14). If Σ<0, we can obtain that
(25)$$\dot{V}\left(t\right)\leq -\lambda_{\min}\left(-\Sigma \right)e^{T}\left(t\right)e\left(t\right)+\kappa$$
when $\lambda_{\min}\left(-\Sigma \right){∥e\left(t\right)∥}^{2}>\kappa$, then it is obvious that $\dot{V}<0$. According to the Lyapunov stability theory, $e\left(t\right)$ is uniformly bounded and converges to a small set. The proof is complete. □

### 2.2. Observer Design for Actuator and Sensor Faults Diagnosis

For the case of simultaneous faults of quadrotor UAV actuator and sensor, the following quadrotor UAV model is established according to Equation (Equation 2), in which not only time delay is incorporated, but also unknown input disturbance and measurement noise disturbance of the UAV during the flight are considered.
(26)$$\left\{\begin{array}{c} \begin{array}{cc} \dot{x}\left(t\right)= & Ax\left(t\right)+A_{z}x\left(t-z\right)+Bu\left(t\right)+g\left(t,x\left(t\right)\right)+g_{z}\left(t,x\left(t-z\right)\right)\\ & +Ef_{a}\left(t\right)+Dd\left(t\right) \end{array}\\ y\left(t\right)=Cx\left(t\right)+Ff_{s}\left(t\right)+D_{1}d_{s}\left(t\right) \end{array}\right.$$
where $x\left(t\right)\in R^{n}$ is the system state vector, $u\left(t\right)\in R^{m}$ donates the system input, $y\left(t\right)\in R^{l}$ is the system output vector. *A*, $A_{z}$, *B*, *C*, *D*, $D_{1}$, *E*, and *F* are real constant matrices of appropriate dimensions. *D*, $D_{1}$, *E*, and *F* are all full column ranks. (A,B) and (A,C) are assumed to be controllable and observable, respectively. $f_{a}\left(t\right)\in R^{r}$ represents the actuator fault, $f_{s}\left(t\right)\in R^{q}$ is sensor fault, $d\left(t\right)\in R^{p}$ represents the unknown input disturbance, $d_{s}\left(t\right)\in R^{s}$ is the measurement noise disturbance, and z∈R is the time delay.

In addition, the following assumptions are introduced.

**Assumption** **4.**
*The sensor fault $f_{s}\left(t\right)$ and disturbance $d\left(t\right)$, $d_{s}\left(t\right)$ are unknown time-varying and satisfy $∥{\dot{f}}_{s}\left(t\right)∥\leq \beta$ with β≥0, $∥\dot{d}\left(t\right)∥\leq \omega$ with ω≥0, $∥{\dot{d}}_{s}\left(t\right)∥\leq \eta$ with η≥0, where β, ω and η are positive numbers.*


To facilitate estimating sensor fault, augmenting the original system (26), a new system is as follows:(27)$$\left\{\begin{array}{c} \begin{array}{cc} {\dot{x}}_{a}\left(t\right)= & A_{a}x_{a}\left(t\right)+A_{za}x_{a}\left(t-z\right)+B_{a}u\left(t\right)+g_{a}+g_{za}+E_{a}f_{a}\left(t\right)\\ & +D_{a}d\left(t\right)+M_{a}\xi \left(t\right) \end{array}\\ y\left(t\right)=C_{a}x_{a}\left(t\right) \end{array}\right.$$
where
$$x_{a}\left(t\right)=\left[\begin{array}{c} \begin{array}{c} \begin{array}{c} x\left(t\right)\\ d_{s}\left(t\right) \end{array} \end{array}\\ f_{s}\left(t\right) \end{array}\right],A_{a}=\left[\begin{array}{ccc} A & 0 & 0\\ 0 & 0 & 0\\ 0 & 0 & 0 \end{array}\right],A_{za}=\left[\begin{array}{ccc} A_{z} & 0 & 0\\ 0 & 0 & 0\\ 0 & 0 & 0 \end{array}\right],B_{a}=\left[\begin{array}{c} B\\ 0\\ 0 \end{array}\right],E_{a}=\left[\begin{array}{c} E\\ 0\\ 0 \end{array}\right],$$
$$D_{a}=\left[\begin{array}{c} D\\ 0\\ 0 \end{array}\right],M_{a}=\left[\begin{array}{cc} 0 & 0\\ I & 0\\ 0 & I \end{array}\right],g_{a}=\left[\begin{array}{c} g\left(t,x\left(t\right)\right)\\ 0\\ 0 \end{array}\right],g_{za}=\left[\begin{array}{c} g_{z}\left(t,x\left(t-z\right)\right)\\ 0\\ 0 \end{array}\right],$$
$$C_{a}=\left[\begin{array}{ccc} C & D_{1} & F \end{array}\right],\xi \left(t\right)=\left[\begin{array}{c} {\dot{d}}_{s}\left(t\right)\\ {\dot{f}}_{s}\left(t\right) \end{array}\right].$$

**Assumption** **5.**
*For every complex number λ with a non-negative real part, the following two equalities hold:*

(28)
$$rank\left[\begin{array}{cc} A_{a}-\lambda I & E_{a}\\ C_{a} & 0 \end{array}\right]=n+q+s+rank\left(E_{a}\right)$$


(29)
$$rank\left[\begin{array}{cc} A_{a}-\lambda I & D_{a}\\ C_{a} & 0 \end{array}\right]=n+q+s+rank\left(D_{a}\right)$$



To design an intermediate observer for the nonlinear system (27), we first introduce two intermediate variables.
(30)$$\tau \left(t\right)=f_{a}\left(t\right)-\mu E_{a}^{T}x_{a}\left(t\right)$$
(31)$$\zeta \left(t\right)=d\left(t\right)-\nu D_{a}^{T}x_{a}\left(t\right)$$

From (27), (30) and (31), we have
(32)$$\begin{array}{cc} \dot{\tau }\left(t\right)= & {\dot{f}}_{a}\left(t\right)-\mu E_{a}^{T}\left[A_{a}x_{a}\left(t\right)+A_{za}x_{a}\left(t-z\right)\right.+g_{a}+g_{za}+B_{a}u\left(t\right)+E_{a}f_{a}\left(t\right)\\ \; & +M_{a}\xi \left(t\right)\left.+D_{a}d\left(t\right)\right] \end{array}$$
(33)$$\begin{array}{cc} \dot{\zeta }\left(t\right)= & \dot{d}\left(t\right)-\nu D_{a}^{T}\left[A_{a}x_{a}\left(t\right)+A_{za}x_{a}\left(t-z\right)\right.+g_{a}+g_{za}+B_{a}u\left(t\right)+E_{a}f_{a}\left(t\right)\\ \; & +M_{a}\xi \left(t\right)\left.+D_{a}d\left(t\right)\right] \end{array}$$

Then, based on (27), (32) and (33), the intermediate estimator is constructed as follows:(34)$$\begin{array}{cc} {\dot{\hat{x}}}_{a}\left(t\right)= & A_{a}{\hat{x}}_{a}\left(t\right)+A_{za}{\hat{x}}_{a}\left(t-z\right)+{\hat{g}}_{a}+{\hat{g}}_{za}+B_{a}u\left(t\right)+E_{a}{\hat{f}}_{a}\left(t\right)\\ \; & +D_{a}\hat{d}\left(t\right)+L\left(y\left(t\right)-\hat{y}\left(t\right)\right) \end{array}$$
(35)$$\begin{array}{cc} \dot{\hat{\tau }}\left(t\right)= & -\mu E_{a}^{T}E_{a}\hat{\tau }\left(t\right)-\mu E_{a}^{T}D_{a}\hat{\zeta }\left(t\right)-\mu E_{a}^{T}\left[A_{a}{\hat{x}}_{a}\left(t\right)+A_{za}{\hat{x}}_{a}\left(t-z\right)+B_{a}u\left(t\right)\right.\\ \; & +{\hat{g}}_{a}+{\hat{g}}_{za}+\left.\mu E_{a}E_{a}^{T}{\hat{x}}_{a}\left(t\right)+\nu D_{a}D_{a}^{T}{\hat{x}}_{a}\left(t\right)\right] \end{array}$$
(36)$$\begin{array}{cc} \dot{\hat{\zeta }}\left(t\right)= & -\nu D_{a}^{T}E_{a}\hat{\tau }\left(t\right)-\nu D_{a}^{T}D_{a}\hat{\zeta }\left(t\right)-\nu D_{a}^{T}\left[A_{a}{\hat{x}}_{a}\left(t\right)+A_{za}{\hat{x}}_{a}\left(t-z\right)+B_{a}u\left(t\right)\right.\\ \; & +{\hat{g}}_{a}+{\hat{g}}_{za}+\left.\mu E_{a}E_{a}^{T}{\hat{x}}_{a}\left(t\right)+\nu D_{a}D_{a}^{T}{\hat{x}}_{a}\left(t\right)\right] \end{array}$$
(37)$$\hat{y}\left(t\right)=C_{a}{\hat{x}}_{a}\left(t\right)$$
(38)$${\hat{f}}_{a}\left(t\right)=\hat{\tau }\left(t\right)+\mu E_{a}^{T}{\hat{x}}_{a}\left(t\right)$$
(39)$$\hat{d}\left(t\right)=\hat{\zeta }\left(t\right)+\nu D_{a}^{T}{\hat{x}}_{a}\left(t\right)$$
(40)$${\hat{d}}_{s}\left(t\right)=\left[\begin{array}{ccc} 0 & I & 0 \end{array}\right]{\hat{x}}_{a}\left(t\right)$$
(41)$${\hat{f}}_{s}\left(t\right)=\left[\begin{array}{ccc} 0 & 0 & I \end{array}\right]{\hat{x}}_{a}\left(t\right)$$
where ${\hat{x}}_{a}\left(t\right)$, $\hat{y}\left(t\right)$, ${\hat{f}}_{a}\left(t\right)$, ${\hat{f}}_{s}\left(t\right)$, $\hat{\tau }\left(t\right)$, $\hat{\zeta }\left(t\right)$, $\hat{d}\left(t\right)$, and ${\hat{d}}_{s}\left(t\right)$ represent the estimation of $x_{a}\left(t\right)$, $y\left(t\right)$, $f_{a}\left(t\right)$, $f_{s}\left(t\right)$, $\tau \left(t\right)$, $\zeta \left(t\right)$, $d\left(t\right)$, and $d_{s}\left(t\right)$, respectively. Define the estimation errors $e_{x_{a}}\left(t\right)=x_{a}\left(t\right)-{\hat{x}}_{a}\left(t\right)$, $e_{x_{a}}\left(t-z\right)=x_{a}\left(t-z\right)-{\hat{x}}_{a}\left(t-z\right)$, $e_{\tau }\left(t\right)=\tau \left(t\right)-\hat{\tau }\left(t\right)$, $e_{\zeta }\left(t\right)=\zeta \left(t\right)-\hat{\zeta }\left(t\right)$, $e_{f_{a}}\left(t\right)=f_{a}\left(t\right)-{\hat{f}}_{a}\left(t\right)$, $e_{d}\left(t\right)=d\left(t\right)-\hat{d}\left(t\right)$, and $e_{d_{s}}\left(t\right)=d_{s}\left(t\right)-{\hat{d}}_{s}\left(t\right)$, then
(42)$$\begin{array}{cc} {\dot{e}}_{x_{a}}\left(t\right)= & \left(A_{a}-LC_{a}+\mu E_{a}E_{a}^{T}+\nu D_{a}D_{a}^{T}\right)e_{x_{a}}\left(t\right)+A_{za}e_{x_{a}}\left(t-z\right)+G_{a}+G_{za}+E_{a}e_{\tau }\left(t\right)\\ \; & +D_{a}e_{\zeta }\left(t\right)+M_{a}\xi \left(t\right) \end{array}$$
(43)$$\begin{array}{cc} {\dot{e}}_{\tau }\left(t\right)= & {\dot{f}}_{a}\left(t\right)-\mu E_{a}^{T}E_{a}e_{\tau }\left(t\right)-\mu E_{a}^{T}D_{a}e_{\zeta }\left(t\right)-\mu E_{a}^{T}\left[\left(A_{a}+\mu E_{a}E_{a}^{T}+\nu D_{a}D_{a}^{T}\right)e_{x_{a}}\left(t\right)\right.\\ \; & +A_{za}e_{x_{a}}\left(t-z\right)\left.+G_{a}+G_{za}+M_{a}\xi \left(t\right)\right] \end{array}$$
(44)$$\begin{array}{cc} {\dot{e}}_{\zeta }\left(t\right)= & \dot{d}\left(t\right)-\nu D_{a}^{T}E_{a}e_{\tau }\left(t\right)-\nu D_{a}^{T}D_{a}e_{\zeta }\left(t\right)-\nu D_{a}^{T}\left[\left(A_{a}+\mu E_{a}E_{a}^{T}+\nu D_{a}D_{a}^{T}\right)e_{x_{a}}\left(t\right)\right.\\ \; & +A_{za}e_{x_{a}}\left(t-z\right)\left.+G_{a}+G_{za}+M_{a}\xi \left(t\right)\right] \end{array}$$
where $G_{a}=g_{a}\left(t,x_{a}\left(t\right)\right)-{\hat{g}}_{a}\left(t,{\hat{x}}_{a}\left(t\right)\right)$, $G_{za}=g_{za}\left(t,x_{a}\left(t-z\right)\right)-{\hat{g}}_{za}\left(t,{\hat{x}}_{a}\left(t-z\right)\right)$.

**Theorem** **2.**
*Suppose the above assumptions hold, for the given positive constants μ, ν, ε, if there exist scalars $\delta_{1}>0$, $\delta_{2}>0$, and matrices P>0, H such that*

(45)
$$\;\varphi =\left[\begin{array}{ccccccccccccc} \varphi_{11} & \varphi_{12} & \varphi_{13} & \varphi_{14} & l_{1}P & PM_{a} & \mu \delta_{1}l_{1}I & \mu \delta_{2}l_{1}I & 0 & 0 & 0 & 0 & 0\\ {}* & \varphi_{22} & \varphi_{23} & \varphi_{24} & 0 & 0 & 0 & 0 & l_{2}P & \mu \delta_{1}l_{2}I & \mu \delta_{2}l_{2}I & 0 & 0\\ {}* & * & \varphi_{33} & \varphi_{34} & 0 & 0 & 0 & 0 & 0 & 0 & 0 & \mu \delta_{1}E_{a}^{T}M_{a} & 0\\ {}* & * & * & \varphi_{44} & 0 & 0 & 0 & 0 & 0 & 0 & 0 & 0 & \mu \delta_{2}D_{a}^{T}M_{a}\\ {}* & * & * & * & -\varepsilon I & 0 & 0 & 0 & 0 & 0 & 0 & 0 & 0\\ {}* & * & * & * & * & -\varepsilon I & 0 & 0 & 0 & 0 & 0 & 0 & 0\\ {}* & * & * & * & * & * & -\varepsilon I & 0 & 0 & 0 & 0 & 0 & 0\\ {}* & * & * & * & * & * & * & -\varepsilon I & 0 & 0 & 0 & 0 & 0\\ {}* & * & * & * & * & * & * & * & -\varepsilon I & 0 & 0 & 0 & 0\\ {}* & * & * & * & * & * & * & * & * & -\varepsilon I & 0 & 0 & 0\\ {}* & * & * & * & * & * & * & * & * & * & -\varepsilon I & 0 & 0\\ {}* & * & * & * & * & * & * & * & * & * & * & -\varepsilon I & 0\\ & * & * & * & * & * & * & * & * & * & * & * & -\varepsilon I \end{array}\right]<0$$

*the estimation error system (41)–(43) is guaranteed to be uniformly ultimately bounded. The estimator gain is given by $L=P^{-1}H$, where $\varphi_{11}=PA_{a}-HC_{a}+A_{a}^{T}P-C_{a}^{T}H^{T}+\mu PE_{a}E_{a}^{T}+\mu E_{a}E_{a}^{T}P+\nu PD_{a}D_{a}^{T}+\nu D_{a}D_{a}^{T}P+2\varepsilon I+Q$, $\varphi_{12}=PA_{za}$, $\varphi_{13}=PE_{a}-\mu \delta_{1}A_{a}^{T}E_{a}-\mu^{2}\delta_{1}E_{a}E_{a}^{T}E_{a}$$-\mu \nu \delta_{1}D_{a}D_{a}^{T}E_{a}$, $\varphi_{14}=PD_{a}-\nu \delta_{2}A_{a}^{T}D_{a}-\nu^{2}\delta_{2}D_{a}D_{a}^{T}D_{a}-\mu \nu \delta_{2}E_{a}E_{a}^{T}D_{a}$, $\varphi_{22}=-Q$, $\varphi_{23}=-\mu \delta_{1}A_{za}^{T}E_{a},\varphi_{24}=-\nu \delta_{2}A_{za}^{T}D_{a}$, $\varphi_{33}=-2\mu \delta_{1}E_{a}^{T}E_{a}+\frac{1}{\varepsilon }I+2\varepsilon E_{a}^{T}E_{a}$, $\varphi_{34}=-\mu \delta_{1}E_{a}^{T}D_{a}-\nu \delta_{2}E_{a}^{T}D_{a}$, $\varphi_{44}=-2\nu \delta_{2}D_{a}^{T}D_{a}+\frac{1}{\varepsilon }I+2\varepsilon D_{a}^{T}D_{a}$.*


**Proof.** Consider the Lyapunov–Krasovskii functional as
(46)$$V=V_{1}+V_{2}+V_{3}+V_{4}$$
where
$$\begin{array}{cc} \; & V_{1}=e_{x_{a}}^{T}\left(t\right)Pe_{x_{a}}\left(t\right),V_{2}=e_{\tau }^{T}\left(t\right)\Gamma_{1}e_{\tau }\left(t\right),V_{3}=e_{\zeta }^{T}\left(t\right)\Gamma_{2}e_{\zeta }\left(t\right),\\ \; & V_{4}=\int_{t-z}^{t}e_{x_{a}}^{T}\left(s\right)Qe_{x_{a}}\left(s\right)ds,\Gamma_{1}=\delta_{1}I,\Gamma_{2}=\delta_{2}I. \end{array}$$Thus
(47)$$\begin{array}{cc} {\dot{V}}_{1}= & e_{x_{a}}^{T}\left(t\right)\left[P\left(A_{a}-LC_{a}+\mu E_{a}E_{a}^{T}+\nu D_{a}D_{a}^{T}\right)\right.+\left.{\left(A_{a}-LC_{a}+\mu E_{a}E_{a}^{T}+\nu D_{a}D_{a}^{T}\right)}^{T}P\right]e_{x_{a}}\left(t\right)\\ \; & +2e_{x_{a}}^{T}\left(t\right)PA_{za}e_{x_{a}}\left(t-z\right)+2e_{x_{a}}^{T}\left(t\right)PG_{a}+2e_{x_{a}}^{T}\left(t\right)PG_{za}+2e_{x_{a}}^{T}\left(t\right)PE_{a}e_{\tau }\left(t\right)\\ \; & +2e_{x_{a}}^{T}\left(t\right)PD_{a}e_{\zeta }\left(t\right)+2e_{x_{a}}^{T}\left(t\right)PM_{a}\xi \left(t\right) \end{array}$$
(48)$$\begin{array}{cc} {\dot{V}}_{2}= & 2\delta_{1}e_{\tau }^{T}\left(t\right){\dot{f}}_{a}\left(t\right)-2\mu \delta_{1}e_{\tau }^{T}\left(t\right)E_{a}^{T}E_{a}e_{\tau }\left(t\right)-2\mu \delta_{1}e_{\tau }^{T}\left(t\right)E_{a}^{T}\left(A_{a}+\mu E_{a}E_{a}^{T}+\nu D_{a}D_{a}^{T}\right)e_{x_{a}}\left(t\right)\\ \; & -2\mu \delta_{1}e_{\tau }^{T}\left(t\right)E_{a}^{T}D_{a}e_{\zeta }\left(t\right)-2\mu \delta_{1}e_{\tau }^{T}\left(t\right)E_{a}^{T}M_{a}\xi \left(t\right)-2\mu \delta_{1}e_{\tau }^{T}\left(t\right)E_{a}^{T}G_{a}-2\mu \delta_{1}e_{\tau }^{T}\left(t\right)E_{a}^{T}G_{za}\\ \; & -2\mu \delta_{1}e_{\tau }^{T}\left(t\right)E_{a}^{T}A_{za}e_{x_{a}}\left(t-z\right) \end{array}$$
(49)$$\begin{array}{cc} {\dot{V}}_{3}= & 2\delta_{2}e_{\zeta }^{T}\left(t\right)\dot{d}\left(t\right)-2\nu \delta_{2}e_{\zeta }^{T}\left(t\right)D_{a}^{T}E_{a}e_{\tau }\left(t\right)-2\nu \delta_{2}e_{\zeta }^{T}\left(t\right)D_{a}^{T}\left(A_{a}+\mu E_{a}E_{a}^{T}+\nu D_{a}D_{a}^{T}\right)e_{x_{a}}\left(t\right)\\ \; & -2\nu \delta_{2}e_{\zeta }^{T}\left(t\right)D_{a}^{T}D_{a}e_{\zeta }\left(t\right)-2\nu \delta_{2}e_{\zeta }^{T}\left(t\right)D_{a}^{T}M_{a}\xi \left(t\right)-2\nu \delta_{2}e_{\zeta }^{T}\left(t\right)D_{a}^{T}G_{a}-2\nu \delta_{2}e_{\zeta }^{T}\left(t\right)D_{a}^{T}G_{za}\\ \; & -2\nu \delta_{2}e_{\zeta }^{T}\left(t\right)D_{a}^{T}A_{za}e_{x_{a}}\left(t-z\right) \end{array}$$
(50)$${\dot{V}}_{4}=e_{x_{a}}^{T}\left(t\right)Qe_{x_{a}}\left(t\right)-e_{x_{a}}^{T}\left(t-z\right)Qe_{x_{a}}\left(t-z\right)$$According to Assumption 4, there exists a scalar γ>0 such that the following equation holds:
(51)$$∥\xi \left(t\right)∥\leq \gamma$$Furthermore, the following inequalities always hold:
(52)$$2e_{x_{a}}^{T}\left(t\right)PG_{a}\leq \frac{1}{\varepsilon }l_{1}^{2}e_{x_{a}}^{T}\left(t\right)P^{2}e_{x_{a}}\left(t\right)+\varepsilon e_{x_{a}}^{T}\left(t\right)e_{x_{a}}\left(t\right)$$
(53)$$2e_{x_{a}}^{T}\left(t\right)PG_{za}\leq \varepsilon e_{x_{a}}^{T}\left(t\right)e_{x_{a}}\left(t\right)+\frac{1}{\varepsilon }l_{2}^{2}e_{x_{a}}^{T}\left(t-z\right)P^{2}e_{x_{a}}\left(t-z\right)$$
(54)$$2e_{x_{a}}^{T}\left(t\right)PM_{a}\xi \left(t\right)\leq \frac{1}{\varepsilon }e_{x_{a}}^{T}\left(t\right)PM_{a}M_{a}^{T}Pe_{x_{a}}\left(t\right)+\varepsilon \gamma^{2}$$
(55)$$2\delta_{1}e_{\tau }^{T}\left(t\right){\dot{f}}_{a}\left(t\right)\leq \frac{1}{\varepsilon }e_{\tau }^{T}\left(t\right)e_{\tau }\left(t\right)+\varepsilon \delta_{1}^{2}\alpha^{2}$$
(56)$$-2\mu \delta_{1}e_{\tau }^{T}\left(t\right)E_{a}^{T}G_{a}\leq \varepsilon e_{\tau }^{T}\left(t\right)E_{a}^{T}E_{a}e_{\tau }\left(t\right)+\frac{1}{\varepsilon }\mu^{2}\delta_{1}^{2}l_{2}^{2}e_{x_{a}}^{T}\left(t\right)e_{x_{a}}\left(t\right)$$
(57)$$-2\mu \delta_{1}e_{\tau }^{T}\left(t\right)E_{a}^{T}G_{za}\leq \varepsilon e_{\tau }^{T}\left(t\right)E_{a}^{T}E_{a}e_{\tau }\left(t\right)+\frac{1}{\varepsilon }\mu^{2}\delta_{1}^{2}l_{2}^{2}e_{x_{a}}^{T}\left(t-z\right)e_{x_{a}}\left(t-z\right)$$
(58)$$-2\mu \delta_{1}e_{\tau }^{T}\left(t\right)E_{a}^{T}M_{a}\xi \left(t\right)\leq \frac{1}{\varepsilon }\mu^{2}\delta_{1}^{2}e_{\tau }^{T}\left(t\right)E_{a}^{T}M_{a}M_{a}^{T}E_{a}e_{\tau }\left(t\right)+\varepsilon \gamma^{2}$$
(59)$$2\delta_{2}e_{\zeta }^{T}\left(t\right)\dot{d}\left(t\right)\leq \frac{1}{\varepsilon }e_{\zeta }^{T}\left(t\right)e_{\zeta }\left(t\right)+\varepsilon \delta_{2}^{2}\omega^{2}$$
(60)$$-2\nu \delta_{2}e_{\zeta }^{T}\left(t\right)D_{a}^{T}G_{a}\leq \varepsilon e_{\zeta }^{T}\left(t\right)D_{a}^{T}D_{a}e_{\zeta }\left(t\right)+\frac{1}{\varepsilon }\nu^{2}\delta_{2}^{2}l_{1}^{2}e_{x_{a}}^{T}\left(t\right)e_{x_{a}}\left(t\right)$$
(61)$$-2\nu \delta_{2}e_{\zeta }^{T}\left(t\right)D_{a}^{T}G_{za}\leq \varepsilon e_{\zeta }^{T}\left(t\right)D_{a}^{T}D_{a}e_{\zeta }\left(t\right)+\frac{1}{\varepsilon }\nu^{2}\delta_{2}^{2}l_{2}^{2}e_{x_{a}}^{T}\left(t-z\right)e_{x_{a}}\left(t-z\right)$$
(62)$$-2\nu \delta_{2}e_{\zeta }^{T}\left(t\right)D_{a}^{T}M_{a}\xi \left(t\right)\leq \frac{1}{\varepsilon }\nu^{2}\delta_{2}^{2}e_{\zeta }^{T}\left(t\right)D_{a}^{T}M_{a}M_{a}^{T}D_{a}e_{\zeta }\left(t\right)+\varepsilon \gamma^{2}$$Define $e\left(t\right)={\left[e_{x_{a}}^{T}\left(t\right)\;e_{x_{a}}^{T}\left(t-z\right)\;e_{\tau }^{T}\left(t\right)\;e_{\zeta }^{T}\left(t\right)\right]}^{T}$, then, we can obtain that
(63)$$\dot{V}=e{\left(t\right)}^{T}\Delta e\left(t\right)+\kappa$$
where
$$\begin{array}{c} \Delta =\left[\begin{array}{cccc} \Delta_{11} & \Delta_{12} & \Delta_{13} & \Delta_{14}\\ {}* & \Delta_{22} & \Delta_{23} & \Delta_{24}\\ {}* & * & \Delta_{33} & \Delta_{34}\\ {}* & * & * & \Delta_{44} \end{array}\right] \end{array}$$
and
$$\begin{array}{cc} \Delta_{11}= & P\left(A_{a}-LC_{a}+\mu E_{a}E_{a}^{T}+\nu D_{a}D_{a}^{T}\right)+{\left(A_{a}-LC_{a}+\mu E_{a}E_{a}^{T}+\nu D_{a}D_{a}^{T}\right)}^{T}P\\ \; & +\frac{1}{\varepsilon }\nu^{2}\delta_{2}^{2}l_{1}^{2}I+2\varepsilon I+Q+\frac{1}{\varepsilon }l_{1}^{2}P^{2}+\frac{1}{\varepsilon }\mu^{2}\delta_{1}^{2}l_{1}^{2}I+\frac{1}{\varepsilon }PM_{a}M_{a}^{T}P \end{array}$$
$$\Delta_{12}=PA_{za}$$
$$\Delta_{13}=PE_{a}-\mu \delta_{1}A_{a}^{T}E_{a}-\mu^{2}\delta_{1}E_{a}E_{a}^{T}E_{a}-\mu \nu \delta_{1}D_{a}D_{a}^{T}E_{a}$$
$$\Delta_{14}=PD_{a}-\nu \delta_{2}A_{a}^{T}D_{a}-\nu^{2}\delta_{2}D_{a}D_{a}^{T}D_{a}-\mu \nu \delta_{2}E_{a}E_{a}^{T}D_{a}$$
$$\Delta_{22}=-Q+\frac{1}{\varepsilon }l_{2}^{2}P^{2}+\frac{1}{\varepsilon }\mu^{2}\delta_{1}^{2}l_{2}^{2}I+\frac{1}{\varepsilon }\nu^{2}\delta_{2}^{2}l_{2}^{2}I$$
$$\Delta_{23}=-\mu \delta_{1}A_{za}^{T}E_{a}$$
$$\Delta_{24}=-\nu \delta_{2}A_{za}^{T}D_{a}$$
$$\Delta_{33}=-2\mu \delta_{1}E_{a}^{T}E_{a}+\frac{1}{\varepsilon }I+2\varepsilon E_{a}^{T}E_{a}+\frac{1}{\varepsilon }\mu^{2}\delta_{1}^{2}E_{a}^{T}M_{a}M_{a}^{T}E_{a}$$
$$\Delta_{34}=-\mu \delta_{1}E_{a}^{T}D_{a}-\nu \delta_{2}E_{a}^{T}D_{a}$$
$$\Delta_{44}=-2\nu \delta_{2}D_{a}^{T}D_{a}+\frac{1}{\varepsilon }I+2\varepsilon D_{a}^{T}D_{a}+\frac{1}{\varepsilon }\nu^{2}\delta_{2}^{2}D_{a}^{T}M_{a}M_{a}^{T}D_{a}$$
$$\kappa =3\varepsilon \gamma^{2}+\varepsilon \delta_{1}^{2}\alpha^{2}+\varepsilon \delta_{2}^{2}\omega^{2}$$According to Schur’s complement theorem, Δ<0 is equivalent to Equation (Equation 44). If Δ<0, we can obtain that
(64)$$\dot{V}\left(t\right)\leq -\lambda_{\min}\left(-\Delta \right)e{\left(t\right)}^{T}e\left(t\right)+\kappa$$
when $\lambda_{\min}\left(-\Delta \right){∥e\left(t\right)∥}^{2}>\kappa$, then it is obvious that $\dot{V}<0$. According to the Lyapunov stability theory, $e\left(t\right)$ is uniformly bounded and converges to a small set. The proof is complete. □

## 3. Simulation Results

The parameters of the QUAV system can be obtained from ref. [18] as shown in Table 1 so that the following matrix can be obtained as the model matrix of the quadrotor UAV system.
$$A=\left[\begin{array}{cccccc} 0 & 1 & 0 & 0 & 0 & 0\\ 0 & 0 & 0 & 0 & 0 & 0\\ 0 & 0 & 0 & 1 & 0 & 0\\ 0 & 0 & 0 & 0 & 0 & 0\\ 0 & 0 & 0 & 0 & 0 & 1\\ 0 & 0 & 0 & 0 & 0 & 0 \end{array}\right],B=\left[\begin{array}{cccc} 0 & 0 & 0 & 0\\ 0 & 0 & 0.4239 & -0.4239\\ 0 & 0 & 0 & 0\\ 0.4239 & -0.4239 & 0 & 0\\ 0 & 0 & 0 & 0\\ -0.0327 & -0.0327 & 0.0327 & 0.0327 \end{array}\right],$$
$$\begin{array}{c} C=\left[\begin{array}{cccccc} 1 & 0 & 0 & 0 & 0 & 0\\ 0 & 0 & 1 & 0 & 0 & 0\\ 0 & 0 & 0 & 0 & 1 & 0 \end{array}\right] \end{array},g\left(t,x\left(t\right)\right)=\left[\begin{array}{c} 0\\ -0.9928x_{4}\left(t\right)x_{6}\left(t\right)-0.1449x_{2}^{2}\left(t\right)\\ 0\\ \;0.9928x_{2}\left(t\right)x_{6}\left(t\right)-0.1449x_{4}^{2}\left(t\right)\\ 0\\ -0.0827x_{6}^{2}\left(t\right) \end{array}\right].$$

The nonlinear term with the time delay is as follows:$$g\left(t,x\left(t-z\right)\right)=\left[\begin{array}{c} 0\\ -2.9784x_{4}\left(t-z\right)x_{6}\left(t-z\right)-0.4347x_{2}^{2}\left(t-z\right)\\ 0\\ 2.9784x_{2}\left(t-z\right)x_{6}\left(t\right)-0.4347x_{4}^{2}\left(t-z\right)\\ 0\\ -0.2481x_{6}^{2}\left(t-z\right) \end{array}\right]$$

### 3.1. Actuator Fault Simulation

Selecting the following matrix $A_{z}$ and actuator fault matrix *E*, the actuator fault is shown in Table 2.
$$A_{z}=\left[\begin{array}{cccccc} 1 & 0.5 & 0 & 0 & 0 & 0\\ 0 & 0 & 0 & 0 & 1 & 0\\ 0 & 0 & -1 & 0.5 & 0 & 0\\ 0 & 1 & 0 & 0 & 0 & 0\\ 0 & 0 & 0 & 0 & 0 & 0.5\\ 0 & 0 & 0 & 0 & -1 & 0 \end{array}\right],$$
$$E={\left[\begin{array}{cccccc} 0 & 1 & 0 & 1 & 0 & -1 \end{array}\right]}^{T}.$$

It can be seen that rank(CE)≠rank(E) and the observer matching condition is not satisfied. Selecting ε=1, σ=1.65, $l_{1}=l_{2}=0.05$, *z* = 0.5 s, 1 s, and 1.5 s. The initial condition is chosen as $x\left(0\right)={\left[0.1,0.1,0.1,0.1,0.1,0.1\right]}^{T}$. Solving (14), it can be obtained that
$$\begin{array}{c} P=10^{3}\left[\begin{array}{cccccc} 7.7580 & -0.1152 & 0.2118 & -0.0057 & -0.5761 & -0.0145\\ -0.1152 & 0.0266 & 0.0096 & -0.0046 & -0.0167 & 0.0197\\ 0.2118 & 0.0096 & 7.3771 & -0.1214 & -0.1600 & -0.0101\\ -0.0057 & -0.0046 & -0.1214 & 0.0162 & -0.0075 & 0.0031\\ -0.5761 & -0.0167 & -0.1600 & -0.0075 & 7.8824 & -0.1182\\ -0.0145 & 0.0197 & -0.0101 & 0.0031 & -0.1182 & 0.0218 \end{array}\right] \end{array},$$
$$\begin{array}{c} H=10^{6}\left[\begin{array}{ccc} 1.1174 & 0.0379 & -0.0754\\ 0.0002 & 0.0001 & -0.0001\\ 0.0379 & 1.0773 & -0.0364\\ -0.0003 & -0.0002 & 0.0003\\ -0.0754 & -0.0364 & 1.1202\\ -0.0001 & -0.0000 & 0.0002 \end{array}\right] \end{array},$$
$$\begin{array}{c} F=10^{3}\left[\begin{array}{ccc} 0.2107 & 0.0299 & -0.0605\\ 5.3939 & 2.3595 & -5.1998\\ 0.0298 & 0.1788 & -0.0291\\ 2.8262 & 2.4802 & -2.7852\\ -0.0612 & -0.0296 & 0.2167\\ -5.4521 & -2.5419 & 6.2169 \end{array}\right] \end{array},$$
$$\begin{array}{c} \delta =1.0594 \end{array}.$$

In the process of system simulation, the choice of parameters is very important. The linear matrix inequalities solved by different σ are different, so the estimation effect will also be different. The choice of σ needs to be adjusted in the simulation process according to the estimation effect. In the process of adjusting the parameters, select an appropriate σ to obtain satisfactory estimation performance.

Figure 1 and Figure 2 show the actuator fault estimation and its estimation error with different time delays. The roll, pitch, yaw angles, and their estimations are shown in Figure 3 and Figure 4, which depict the estimation errors of states. From the simulation results, the method proposed in this paper can achieve a relatively accurate fault estimation effect, and its estimation error is within an acceptable range. It can be seen from Figure 3 and Figure 4 that the method proposed in this paper can accurately estimate the system state when the time delay *z* is 1 s.

### 3.2. Actuator and Sensor Faults Simulation

If choosing the output vector $y\left(t\right)={\left[\phi \;\dot{\phi }\;\theta \;\dot{\theta }\;\psi \;\dot{\psi }\right]}^{T}$, the matrix *C* can be obtained as $$C=\left[\begin{array}{cccccc} 1 & 0 & 0 & 0 & 0 & 0\\ 0 & 1 & 0 & 0 & 0 & 0\\ 0 & 0 & 1 & 0 & 0 & 0\\ 0 & 0 & 0 & 1 & 0 & 0\\ 0 & 0 & 0 & 0 & 1 & 0\\ 0 & 0 & 0 & 0 & 0 & 1 \end{array}\right]$$

Selecting the following matrix $A_{z}$, the actuator fault matrix *E*, sensor fault matrix *F*, unknown input disturbance matrix *D*, and measurement noise disturbance matrix $D_{1}$.
$$A_{z}=\left[\begin{array}{cccccc} 0 & 0.1 & 0 & 0 & 0 & 0\\ 0 & 0 & 0 & 0 & 0 & 0\\ 0 & 0 & 0 & 0.1 & 0 & 0\\ 0 & 0 & 0 & 0 & 0 & 0\\ 0 & 0 & 0 & 0 & 0 & 0.1\\ 0 & 0 & 0 & 0 & 0 & 0 \end{array}\right],$$
$$\begin{array}{cc} \; & D={\left[\begin{array}{cccccc} -2 & 0 & -1 & 0 & 2 & 2 \end{array}\right]}^{T},\\ \; & D_{1}={\left[\begin{array}{cccccc} -1 & 0 & 1 & 0 & 2 & 3 \end{array}\right]}^{T},\\ \; & E={\left[\begin{array}{cccccc} 2.5 & -2.8 & 0 & -3 & 0 & 0 \end{array}\right]}^{T},\\ \; & F={\left[\begin{array}{cccccc} 0 & 1 & -1 & 1 & -1 & 1 \end{array}\right]}^{T}. \end{array}$$

The actuator fault $f_{a}\left(t\right)$, sensor fault $f_{s}\left(t\right)$, unknown input disturbance $d\left(t\right)$, and measurement noise disturbance $d_{s}\left(t\right)$ are created as Table 3.

Selecting ε=1, μ=0.42, ν=0.52, $l_{1}=l_{2}=0.01$, *z* = 0.5 s, *z* = 1 s, and *z* = 1.5 s, the initial condition is chosen as $x\left(0\right)={\left[0.5,0,0.5,0,0.5,0\right]}^{T}$, and solving (45), it can be obtained that
$$\;P=\left[\begin{array}{cccccccc} 140.0250 & 25.0471 & -88.2987 & 23.9292 & 69.1597 & -25.0357 & -0.7087 & -0.0943\\ 25.0471 & 142.5372 & -41.5446 & -80.4350 & 43.5560 & -43.1932 & 0.0680 & 1.8197\\ -88.2987 & -41.5446 & 624.5689 & -30.9302 & -164.9707 & 304.4232 & 0.7686 & -1.3753\\ 23.9292 & -80.4350 & -30.9302 & 135.9678 & 46.8234 & -42.0977 & 0.0817 & 1.8518\\ 69.1597 & 43.5560 & -164.9707 & 46.8234 & 359.3018 & -350.8734 & 1.2579 & -1.6966\\ -25.0357 & -43.1932 & 304.4232 & -42.0977 & -350.8734 & 445.8583 & 1.7296 & 1.7615\\ -0.7087 & 0.0680 & 0.7686 & 0.0817 & 1.2579 & 1.7296 & 9.3141 & 0.0717\\ -0.0943 & 1.8197 & -1.3753 & 1.8518 & -1.6966 & 1.7615 & 0.0717 & 8.8824 \end{array}\right],$$
$$H=\left[\begin{array}{cccccc} 567.6411 & -76.0531 & 99.8895 & -258.8584 & -144.7848 & 231.0494\\ -123.2841 & 368.6634 & 31.3012 & 86.9136 & 164.8559 & -159.8886\\ 92.4390 & -30.0453 & 411.7196 & 387.5852 & -95.5340 & -45.2390\\ -216.3821 & 118.7546 & 330.7784 & 366.7477 & -50.8788 & -116.5869\\ -54.3820 & 76.0824 & -214.5299 & 67.1050 & 264.2724 & -90.9609\\ 38.3697 & -30.2844 & -121.2659 & 241.5669 & 255.3811 & -121.6789\\ 21.4441 & -7.5532 & 37.9477 & -96.8781 & -20.8478 & 108.4977\\ 46.9471 & -3.9202 & -77.4701 & 6.5439 & -89.1012 & 107.0822 \end{array}\right],$$
$$L=\left[\begin{array}{cccccc} 10.5190 & -3.4548 & 6.5607 & -7.9338 & -9.9115 & 7.6486\\ -5.7107 & 6.3437 & 3.9050 & 5.7067 & 2.5811 & -4.6508\\ 3.2315 & -0.7418 & 4.8338 & -2.1402 & -5.1619 & 2.8658\\ -6.2555 & 5.3596 & 5.4766 & 7.0649 & 1.5457 & -4.5197\\ -7.9011 & 1.7818 & -13.6159 & 8.3346 & 15.7623 & -8.4974\\ -8.9470 & 2.7872 & -13.0173 & 9.3998 & 16.4181 & -9.4799\\ 5.5950 & -1.8395 & 8.4305 & -13.7840 & -7.6844 & 14.8634\\ 8.5915 & -3.2074 & -9.9326 & -2.4818 & -11.9699 & 14.6125 \end{array}\right],$$
$$\delta_{1}=15.4716,\delta_{2}=13.5147.$$

When the actuator and sensor of the quadrotor UAV system fail at the same time, considering the influence of time delay and disturbance, in order to obtain the estimated value of the fault and disturbance, two intermediate variables are introduced. Therefore, compared with the single actuator fault diagnosis, there is one more parameter here, and the selection of μ and ν should be adjusted during the simulation process to obtain a better estimation effect.

Through system simulation, when the time delays are 0.5 s, 1 s, and 1.5 s, respectively, the actuator fault $f_{a}\left(t\right)$, sensor fault $f_{s}\left(t\right)$, unknown input disturbance $d\left(t\right)$, measurement noise disturbance $d\left(t\right)$, and their estimated values ${\hat{f}}_{a}\left(t\right)$, ${\hat{f}}_{s}\left(t\right)$, $\hat{d}\left(t\right)$, and $\hat{d_{s}}\left(t\right)$ are shown in Figure 5, Figure 6, Figure 7 and Figure 8. Figure 9 presents the estimated errors for faults and disturbances, and the system states estimations are shown in Figure 10. Figure 11 depicts the estimation errors of states.

From the simulation results, it can be seen that the algorithm can accurately estimate actuator fault, sensor fault, unknown input disturbance, and measurement noise disturbance under different time delays. From the estimation error curve, when the time delay is 1 s, the estimation errors of actuator fault, sensor fault, and disturbance are very small, all within the acceptable range. The roll angle, pitch angle, yaw angle, and their estimates are shown in Figure 10. The simulation results show that the observer method proposed in this paper can achieve good state estimation.

## 4. Conclusions

In this paper, a fault estimation observer is proposed for the nonlinear system of QUAV with time delay. Firstly, for a single actuator fault system, an intermediate variable is introduced, an intermediate observer is designed for the system, and the estimated value of the actuator fault can be obtained. Secondly, the influence of disturbances is considered, when the actuator and sensor fail at the same time, the original system is augmented, and then two intermediate variables are introduced to design a fault estimator for the augmented system, and the estimated value of the actuator fault, sensor fault, and disturbances can be obtained. Through system simulation results, the feasibility and effectiveness of the algorithm are verified. For the system, fault diagnosis is very important. On this basis, an appropriate FTC strategy should also be designed to compensate for the fault and maintain the stability of the system. Therefore, future work will consider the FTC of QUAV systems based on fault estimation results.

## Figures and Tables

**Figure 1 sensors-23-00034-f001:**
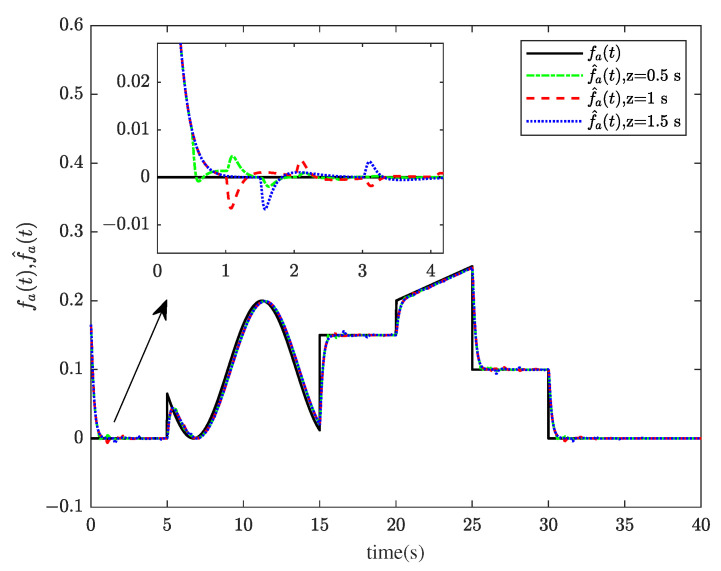
The actuator fault $f_{a}\left(t\right)$ and its estimation ${\hat{f}}_{a}\left(t\right)$.

**Figure 2 sensors-23-00034-f002:**
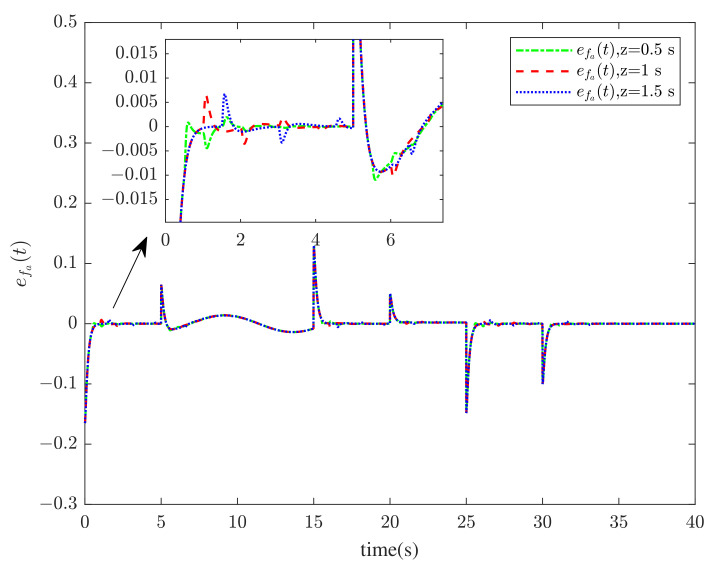
The actuator fault estimation errors $e_{f_{a}}\left(t\right)$ when *z* = 0.5 s, 1 s, 1.5 s.

**Figure 3 sensors-23-00034-f003:**
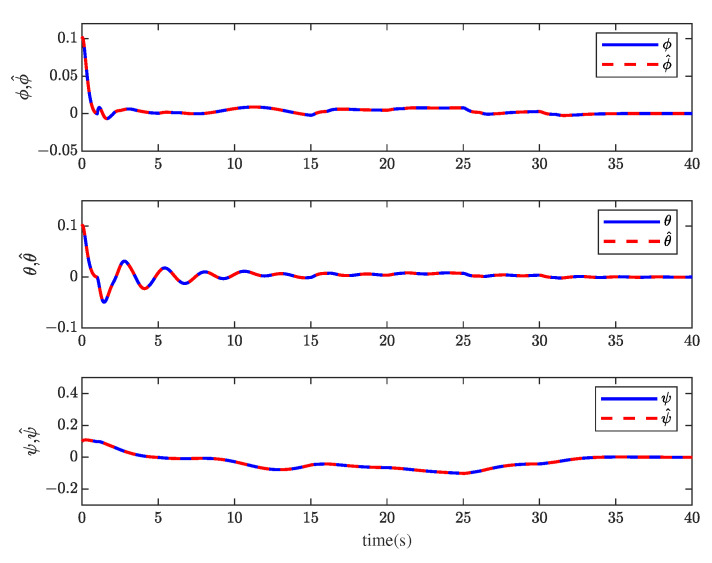
The roll angle ϕ, pitch angle θ, yaw angle ψ, and their estimation $\hat{\phi }$, $\hat{\theta }$ and $\hat{\psi }$ when *z* = 1 s.

**Figure 4 sensors-23-00034-f004:**
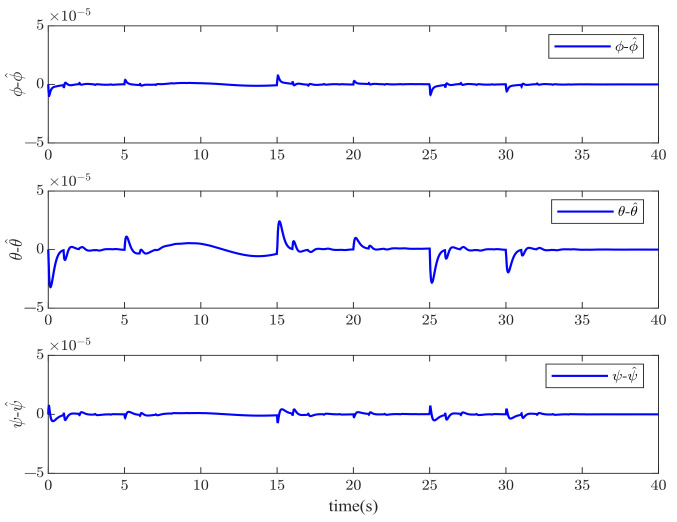
The estimation errors of roll angle ϕ, pitch angle θ, and yaw angle ψ when *z* = 1 s.

**Figure 5 sensors-23-00034-f005:**
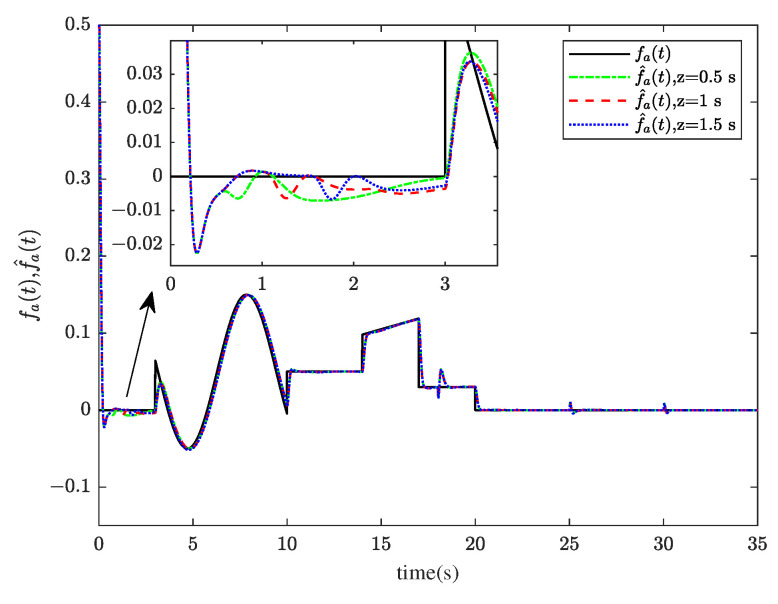
The actuator fault $f_{a}\left(t\right)$ and its estimation ${\hat{f}}_{a}\left(t\right)$.

**Figure 6 sensors-23-00034-f006:**
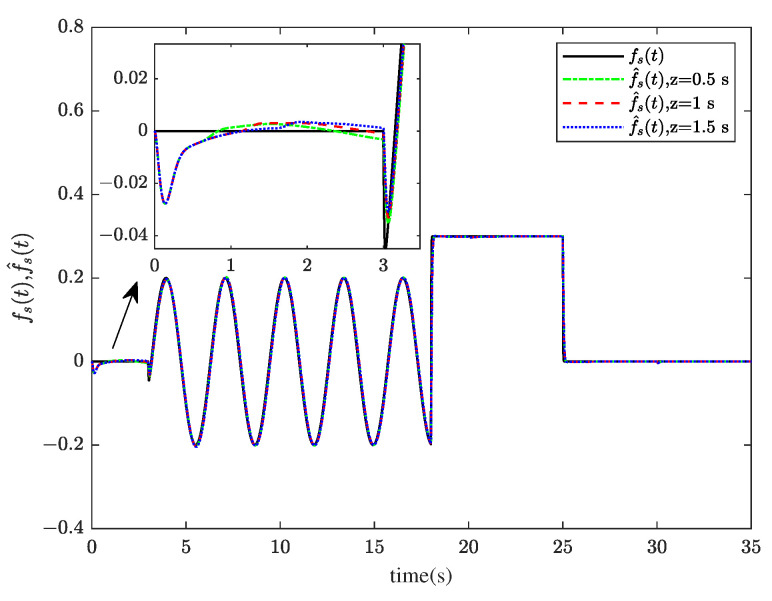
The sensor fault $f_{s}\left(t\right)$ and its estimation ${\hat{f}}_{s}\left(t\right)$.

**Figure 7 sensors-23-00034-f007:**
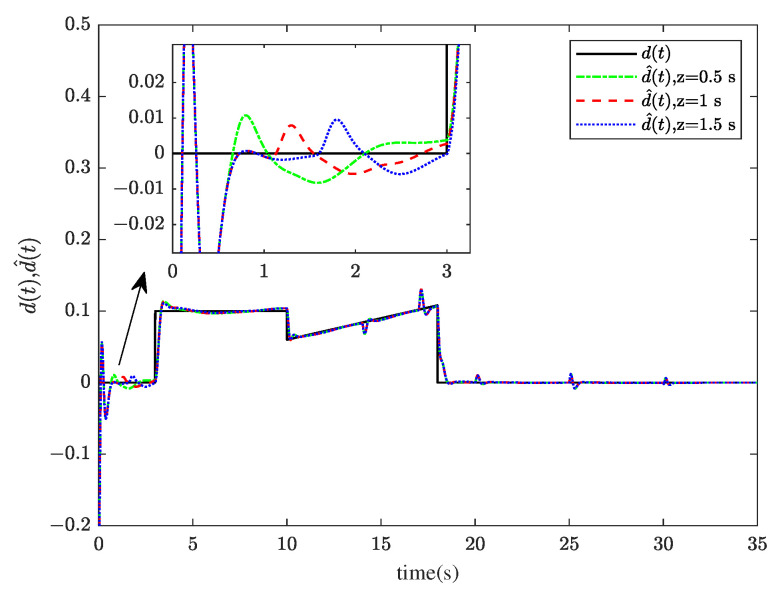
The unknown input disturbance $d\left(t\right)$ and its estimation $\hat{d}\left(t\right)$.

**Figure 8 sensors-23-00034-f008:**
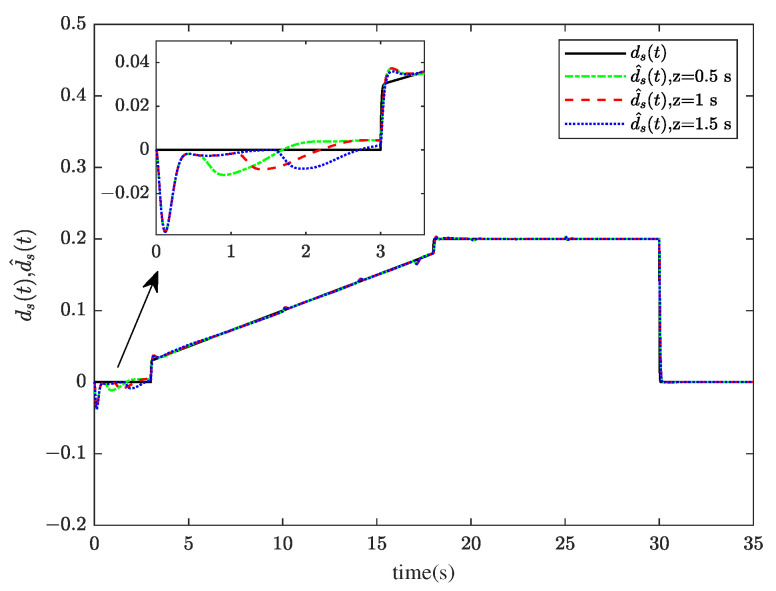
The measurement noise disturbance $d_{s}\left(t\right)$ and its estimation $\hat{d_{s}}\left(t\right)$.

**Figure 9 sensors-23-00034-f009:**
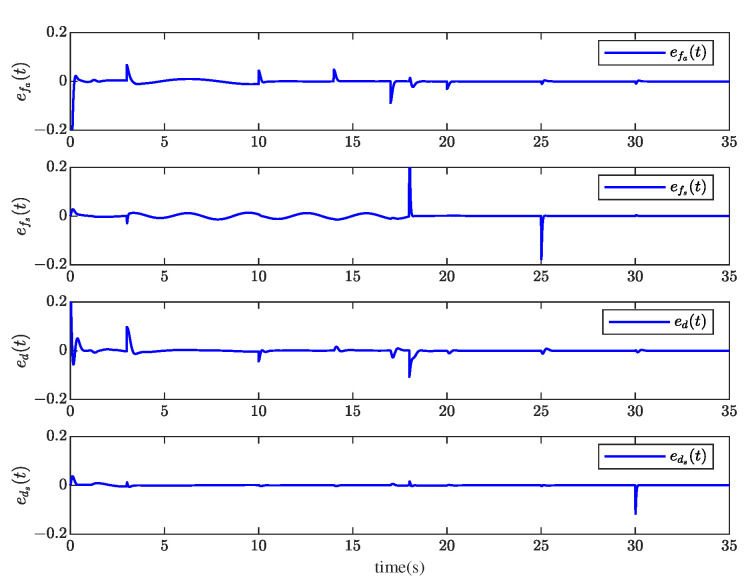
The actuator fault, sensor fault, and disturbance estimation errors $e_{f_{a}}\left(t\right)$, $e_{f_{s}}\left(t\right)$, $e_{d}\left(t\right)$, and $e_{d_{s}}\left(t\right)$ when *z* = 1 s.

**Figure 10 sensors-23-00034-f010:**
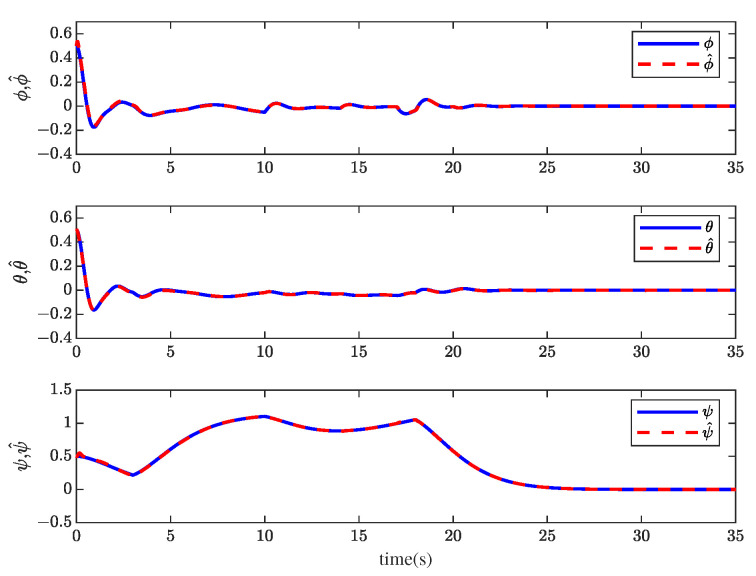
The roll angle ϕ, pitch angle θ, yaw angle ψ, and their estimation $\hat{\phi }$, $\hat{\theta }$, and $\hat{\psi }$ when *z* = 1 s.

**Figure 11 sensors-23-00034-f011:**
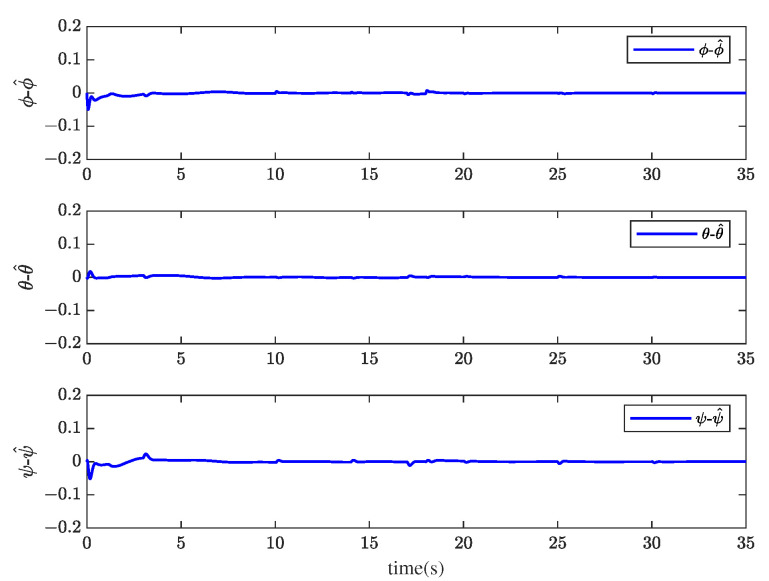
The estimation errors of roll angle ϕ, pitch angle θ, and yaw angle ψ when *z* = 1 s.

**Table 1 sensors-23-00034-t001:** QUAV system parameters.

Parameters	Value	Unit
$K_{vc}$	−0.0036	N·m/V
$K_{vn}$	0.0036	N·m/V
$K_{afcx}$	0.008	N·m·s${}^{2}$
$K_{afcy}$	0.008	N·m·s${}^{2}$
$K_{afcz}$	0.0091	N·m·s${}^{2}$
$K_{lc}$	0.1188	N·m/V
$J_{x}$	0.0552	kg·m${}^{2}$
$J_{y}$	0.0552	kg·m${}^{2}$
$J_{z}$	0.110	kg·m${}^{2}$
*l*	0.197	m

**Table 2 sensors-23-00034-t002:** Actuator fault $f_{a}\left(t\right)$.

*t*	[0 s, 5 s)	[5 s, 20 s)	[15 s, 20 s)	[20 s, 25 s)	[25 s, 30 s)	[30 s, 40 s]
$f_{a}\left(t\right)$	0	0.1+0.1sin(0.7t)	0.15	0.01t	0.1	0

**Table 3 sensors-23-00034-t003:** Actuator fault $f_{a}\left(t\right)$, sensor fault $f_{s}\left(t\right)$, disturbance $d\left(t\right)$ and $d_{s}\left(t\right)$.

*t*	[0 s, 3 s)	[3 s, 10 s)	[10 s, 14 s)	[14 s, 17 s)	[17 s, 20 s)	[20 s, 35 s]
$f_{a}\left(t\right)$	0	0.05+0.1sint	0.05	0.007t	0.03	0
** t **	** [0 s, 3 s)**	** [3 s, 18 s)**	** [18 s, 25 s)**	** [25 s, 35 s]**		
$f_{s}\left(t\right)$	0	0.2sin(2t)	0.3	0		
** t **	** [0 s, 3 s)**	** [3 s, 10 s)**	** [10 s, 18 s)**	** [18 s, 35 s)**		
$d\left(t\right)$	0	0.1	0.006t	0		
** t **	** [0 s, 3 s)**	** [3 s, 18 s)**	** [18 s, 30 s)**	** [30 s, 35 s)**		
$d_{s}\left(t\right)$	0	0.01t	0.2	0		

## Data Availability

Not applicable.

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
