# Peer review of "Fault Estimation Method for Nonlinear Time-Delay System Based on Intermediate Observer-Application on Quadrotor Unmanned Aerial Vehicle"

_sensors, 2022, doi:10.3390/s23010034_

Round 1

Reviewer 1 Report

The paper focuses on actuator and sensor fault estimation for quadrotor UAVs. The fault estimation is done based on an observer which is designed via Lyapunov-Krasovskii functional. The paper is well written and organized, however the simulation results need further clarification. My specific comments are the following:

1. There is no noise on the measured signals. Please add sensor typical sensor noise, without these the simulation is oversimplified.

2. How fast does the observer converge to the actual states? Since the observer is initialized with the actual state values it is impossible to see this.

3. How sensitive is the designed observer to model uncertainties? What happens if the QUAV parameters are slightly off from the nominal values?

I recommend extending the paper with the details mentioned above and resubmitting the revised version.

Reviewer 2 Report

Comments on Manuscript sensors-2019556

In this paper, the actuator and sensor faults estimation issues are discussed for a quadrotor unmanned aerial vehicle (QUAV) system is with time-delay and nonlinear term. For simultaneous actuator and sensor faults, the system is first augmented, and then two intermediate variables are introduced. One is for estimating the actuator fault, and the other is for dealing with the external disturbance. Consequently, an augmented observer is designed for estimating the system state, the actuator fault, the sensor fault and the disturbance simultaneously. The topic is interesting. I have the following comments.

(1)   Assumption 3 is the minim phase condition. Then, What the function of this assumptions? Or, where is it used. The authors should give some explanations.

(2)   In (6), an intermediate variable is constructed. It is constructed based on the actuator and the system state variable. Then, why do you construct intermediate variable in this way?

(3)   I suppose that the propose method can only be applied on those systems where the dimension of the outputs are lager than or equals to the total dimensions of all the unknown information: the actuator fault, the sensor fault and the external disturbance. This is actual a restrict of the method. The authors should give some explanations to this issue.

(4)   In the simulation part, the state estimation performance should also be given.

(5)   Some new reference should be observed by the authors: 

Interval-observer-based fault detection and isolation design for T-S fuzzy system based on zonotope analysis, IEEE Transactions on Fuzzy Systems

Round 2

Reviewer 1 Report

The authors revised the paper based on my comments on the initial version of the paper. Therefore, I recommend accepting the paper as is.

Reviewer 2 Report

Although the authors gave the responses to my concerns, some of them are not satisfied! Therefore, some further revisions are need for the authors to complete. 1.    1.  In my pervious comments, I asked “What the function of this assumptions? Or, where is it used.” The authors did not give the reasonable answer! Firstly, the authors did not response the question of “where is it used”. In your designs or formula derivation, how can you use it? Or what’s its function exactly. The authors did not give the answer properly.

2. I do not think that this paper provides a way to break through the restriction of the observer matching condition as claimed by the authors in Remark 1. In Remark 1, the authors claimed that “The intermediate observer used in this paper only needs to satisfy the condition that E is full column rank, which overcomes the restriction of the observer matching condition rank(CE) = rank(E).” However, in the simulation section, the example provided by the authors satisfies the observer matching condition. I am very sure, without this condition, the unknown input reconstruction cannot be reached! If the authors persist in their statement, an example which can support their statement should be given.

3. There are still grammar mistakes needed to be improved. For example, the sentence in page “……but the system does not consider the influence of external disturbance and other factors” the “and” should be “or”. Thus, the author should check the manuscript very carefully to correct all the grammar mistakes.
